# Bayesian Analysis of the Effects of Olive Oil-Derived Antioxidants on Cryopreserved Buck Sperm Parameters

**DOI:** 10.3390/ani11072032

**Published:** 2021-07-07

**Authors:** Ander Arando Arbulu, Francisco Javier Navas González, Alejandra Bermúdez-Oria, Juan Vicente Delgado Bermejo, África Fernández-Prior, Antonio González Ariza, Jose Manuel León Jurado, Carlos Carmelo Pérez-Marín

**Affiliations:** 1Department of Genetics, Faculty of Veterinary Medicine, University of Cordoba, 14014 Cordoba, Spain; anderarando@hotmail.com (A.A.A.); juanviagr218@gmail.co (J.V.D.B.); angoarvet@outlook.es (A.G.A.); 2Animal Breeding Consulting, S.L., University of Cordoba, 14014 Cordoba, Spain; 3Institute of Agricultural Research and Training (IFAPA), Alameda del Obispo, 14004 Cordoba, Spain; 4Instituto de la Grasa, Consejo Superior de Investigaciones Científicas (CSIC), 41013 Sevilla, Spain; aleberori@ig.csic.es (A.B.-O.); mafprior@ig.csic.es (Á.F.-P.); 5Centro Agropecuario Provincial de la Diputación de Cordoba, 14014 Cordoba, Spain; jomalejur@yahoo.es; 6Department of Animal Medicine and Surgery, Faculty of Veterinary Medicine, University of Cordoba, 14014 Cordoba, Spain; pv2pemac@uco.es

**Keywords:** phenolic antioxidant, olive oil, caprine, spermatozoa, Bayesian inference

## Abstract

**Simple Summary:**

The use of olive oil by-products for caprine sperm cryopreservation offers an interesting opportunity to improve post-thawed sperm quality, as antioxidants such as hydroxytyrosol (HT) and 3,4-dihydroxyphenylglycol (DHPG) could reduce lipid peroxidation. Therefore, this study provides evidence of the positive effect of the addition of HT, DHPG, or the mixture of both antioxidants in cryopreserved buck sperm. In addition, the application of Bayesian statistics for data analysis may enable quantifying the dimensionality of the real effect of antioxidants on sperm.

**Abstract:**

The present study evaluates the effect of olive oil-derived antioxidants, hydroxytyrosol (HT) and 3,4-dihydroxyphenylglycol (DHPG), on cryopreserved caprine sperm using Bayesian inference of ANOVA. For this proposal, sperm was collected, pooled and diluted in freezing media supplemented with different concentrations of HT, DHPG and the mixture (MIX) of both antioxidants. Sperm motility, viability, acrosome integrity, mitochondrial status, and lipid peroxidation (LPO) were assessed in fresh and frozen-thawed sperm samples. The results provided evidence that HT at low concentrations improves sperm motility and viability, and reduces the LPO. Contrastingly, DHPG and MIX exert a positive effect by reducing LPO values as concentration increases. Additionally, mitochondrial potential was reduced when samples were supplemented with HT at low concentrations and mixture of both antioxidants. Conclusively, the addition of olive oil-derived antioxidants (HT at 10 µg/mL and DHPG at 30 µg/mL) implements a protective effect in cryopreserved buck sperm. Bayesian analysis alternatives offer new possibilities to determine the repercussion of antioxidants on sperm, both quantitatively and qualitatively.

## 1. Introduction

For the goat industry, the combination of artificial insemination and sperm cryopreservation is an optimal manner in which to speed up genetic improvement while reducing the incidence of sexually transmitted diseases. However, sperm cryopreservation procedures are associated with irreversible damage in sperm cells, which compromises sperm fertility due to of cold shock, osmotic stress, and intracellular ice crystals formation, among others [1]. One of the reasons of the vulnerability of goat spermatozoa to freezing-thawing procedures is the composition of their plasma membrane, which contains large amounts of polyunsaturated fatty acids [2]. As a consequence, such sperm turns highly susceptible to membrane peroxidation derived from the lipid oxidation of membrane by reactive oxygen species (ROS) [3]. The presence of prooxidant molecules, such as free radicals and ROS, is strongly related to metabolic stress and spermatozoa damage during cryopreservation [4].

Nevertheless, ROS have a variable effect on spermatozoa and their impact hinges on the nature and concentration of these substances. When in physiological concentrations, a promoting effect has been reported on capacitation, acrosomal reaction, and sperm zona pellucida interactions [5]. By contrast, high concentrations of ROS are related to spermatozoa normal function inhibition, thus the reduction of sperm viability due to the oxidative stress (OE) and the subsequent peroxidation of polyunsaturated fatty acids in their membranes [6].

Under normal conditions, spermatozoa have endogenous mechanisms to deal with OE. In the sperm plasma, enzymatic and non-enzymatic antioxidants are present, and they participate in the balance mechanism to prevent OE [7]. By contrast, when spermatozoa are exposed to stress conditions, endogenous antioxidants cannot counteract the excess of free radicals; thus, the addition of exogenous antioxidants may be essential to preserve the quality of these cells. In this sense, several studies have evaluated the use of exogenous antioxidants in goat sperm [3,8,9,10,11]. In this framework, a large number of natural compounds has been tested in cell cultures assessing the antioxidant, anti-inflammatory or chelating properties of antioxidants, in recent years, e.g., mentha [12], *Feijoa sellowiana* [13], grapes [14], or olives [15].

Olive fruits (*Olea europea*), olive oil, and its derivates present a large amount of phenolic components, for which remarkable antioxidant properties have been reported [16], with corroborated advantages for human health [17]. Two of the most important phenolic compounds present in olive fruit are hydroxytyrosol (3,4 dyhydroxyphenylethanol, HT) and 3,4-dihydroxyphenylglycol (DHPG), which are isolated from the alperujo olive pulp (semi-solid waste generated in the two-phase system used in olive oil extraction). HT is a simple phenol with significant antioxidant properties [16], which reduces the oxidation of low-density lipoproteins, protect against H_2_O_2_ cytotoxicity and minimize lactate dehydrogenase activity [18,19,20]. Regarding DHPG, powerful antioxidant and potential anti-inflammatory effects have been reported which even compare to those reported for vitamin E [21]. 

The use of HT and DHPH is widespread in human health studies. By contrast, there are few studies on the effects derived from the addition of these antioxidants to sperm dilution media in animals. Contextually, HT-supplemented sperm extender has previously been evaluated in studies conducted on rats [22], humans [23], and rams [15,24,25]. However, the effect of DHPG supplementation has only been reported in ram sperm [15,25]. Taking into account the properties of both antioxidants, the present study hypothesizes that extenders supplemented with these compounds might counteract the sperm damage inflicted by the cryopreservation process. Thus, the aim of the present study was to evaluate the effect of freezing extenders supplemented by different concentrations of HT, DHPG, and the mixture of both substances on the post-thawed sperm quality of goat semen. The effects of the increasing concentrations on each sperm parameter pair correlation were studied.

## 2. Materials and Methods

### 2.1. Chemicals

HT and DHPG, stock solution 76.9 mM and 14.7 mM, respectively (Figure 1), were extracted and purified from olive by-products (alperujo) following the processes described by Fernandez-Bolaños et al. [26] and Fernández-Bolaños Guzmán et al. [27]. A commercial TRIS-based extender (Biladyl, Minitube Iberica, Tarragona, Spain) was used to centrifuge and freeze sperm. LIVE/DEAD^®^ sperm viability kit, composed by SYBR-14 and propidium iodide (PI), Mitotracker Red CMXRos and C_11_-BODIPY^581/591^ were purchased from Molecular Probes Europe (Leiden, The Netherlands). Peanut agglutinin conjugated with fluorescein isothiocyanate (PNA-FITC) was obtained in Sigma-Aldrich (St. Louis, MO, USA).

### 2.2. Animals and Semen Collection

Semen was collected from six Murciano-Granadina breed bucks (4–5 years old). The animals involved in the study were located at the Centro Agropecuario Provincial de Córdoba (Córdoba, Spain) and were managed following the prescriptions and regulations of the European Union (2010/63/EU) in its transposition to Spanish law (RD 53/2013). A total of 12 ejaculates per animal (72 ejaculates in total) were collected with an artificial vagina twice a week during the non-breeding season. Previously, semen had been collected for one month in order to ensure the renewal of the epididymal reserves.

After collection, the ejaculates were placed in a water bath at 37 °C during evaluation and they were assessed to determine volume by graduated tubes, sperm concentration by photometer (Accurread, IMV technologies, France) and mass motility, by placing 5 μL of raw semen on a preheated slide (37 °C) and observed in the optical microscope (40× magnification; Olympus, Tokyo, Japan). Sperm mass motility was scored subjectively from 0 (no motile spermatozoa) to 5 (numerous rapid waves) as described by Evans and Maxwell [28] and Lopes et al. [29]. The inclusive criteria for ejaculates to be considered in the study were: volume ≥ 0.5 mL, concentration ≥ 3000 × 10^6^ spz/mL and masal motility ≥ 4 (ejaculates with more than 70% total motility).

### 2.3. Experimental Design

As schematized in Figure 2, every sperm collection day, sperm samples were splitted into 13 different aliquots and diluted with extenders containing different (or null) concentrations of HT, DHPG, or a mixture of both antioxidants (MIX), to obtain the final concentrations as follows: Control (without antioxidant); HT1 (10 μg/mL); HT2 (30 μg/mL); HT3 (50 μg/mL) and HT4 (70 μg/mL); DHPG1 (10 μg/mL); DHPG2 (30 μg/mL); DHPG3 (50 μg/mL); DHPG4 (70 μg/mL); MIX1 (5 μg/mL HT + 5 μg/mL DHPG); MIX2 (15 μg/mL HT + 15 μg/mL DHPG); MIX3 (25 μg/mL HT + 25 μg/mL DHPG); MIX4 (35 μg/mL HT + 35 μg/mL DHPG). In order to maintain the aforementioned final antioxidant concentrations, the fact that the extender used (Biladyl) requires a two-step process was considered. Therefore, the same antioxidant concentration was added to both fractions used (FAey and FBey), as described below. 

Sperm motility, viability, acrosome integrity, mitochondrial membrane potential (HMMP), and membrane lipid peroxidation (LPO) were assessed in frozen-thawed sperm samples. Twelve repetitions of the experiment were performed.

### 2.4. Semen Dilution and Freezing 

After collection, ejaculates were diluted at 1:2 with TRIS-based extender (FA of Biladyl without egg yolk) for individual evaluation and, if inclusion criteria were reached, these were pooled and diluted to reach a dilution of 1:10. To remove seminal plasma, pooled samples were centrifuged at 600× *g* for 15 min. The supernatant was removed and the pellet was resuspended, adding a volume of fraction A of Biladyl containing egg yolk (FAey). Then, the pooled sample was split into 13 different aliquots; 12 samples were prepared by adding FAey supplemented with the previously described HT, DHPG and MIX antioxidant concentrations. A control group (no antioxidant) was also prepared. The samples were then immediately placed in a programmable freezer (cell incubator SH-020S, Welson, Korea) to reach 5 °C and maintained for two hours at 5 °C.

Then, each aliquot was diluted with Tris-egg yolk-glycerol extender (FBey) supplemented with HT, DHPG and/or MIX, obtaining a final volume of 1000 µL per sample (with a concentration of 400 × 10^6^ spz/mL). The samples were loaded into 0.25 mL straws (100 × 10^6^ spz/straws) and maintained for two hours at 5 °C.

Three straws per sample were frozen using liquid nitrogen vapors. Straws were horizontally placed in racks 4 cm above the liquid nitrogen level for 10 min and then plunged in liquid nitrogen pending analysis. For thawing, the samples were immersed in a water bath at 37 °C for 30 s.

### 2.5. Sperm Quality Assessment

#### 2.5.1. Motility

ISAS software v.1.2 (Integrated Semen Analyser System, Proiser, Valencia, Spain) equipped with an HS640C video camera was used to assess sperm motility. Sperm samples were diluted in FAey at a final concentration of 25 × 10^6^ spz/mL and, after 10 min incubation, 5 µL of each diluted sample was evaluated using a slide and covered (22 × 22 mm). Four fields and a minimum of 500 spermatozoa were randomly captured at 10× magnification using a UB203i phase contrast microscope (Chongqing UOP Photoelectric Technology Co., Ltd, Beibei District, Chongqing, China). A total of 25 images per second were acquired, selecting particles with an area of between 10 and 70 μm^2^ and categorized as motile when VAP >10 μm/s, and linearly motile when they deviated >75% from a straight line. The analyses provided information about total motility (TM, %) and progressive motility (PM, %), curvilinear velocity (VCL, μm/s), straight line velocity (VSL, μm/s), average path velocity (VAP, μm/s), straightness (STR, %), linearity (LIN, %), wobble (WOB, %), amplitude of lateral head displacement (ALH, μm), and beat/cross frequency (BCF, Hz).

#### 2.5.2. Flow Cytometer

The recommendations of the International Society for Advancement of Cytometry were followed to perform flow cytometric analyses [30] using a FACScalibur flow cytometer (BD Biosciences, San Jose, CA, USA) equipped with a 488 nm argon blue laser. Sheath flow rate was set at 12.0 ± 3 µL/min in all analyses (LOW mode). Green fluorescence from SYBR-14, PNA-FITC and C_11_-BODIPY^581/591^ was read with an FL1 photodetector (530/30 band-pass filter). Red fluorescence PI and Mitotracker Red CMXRos was read with an FL2 photodetector (585/42 nm bandpass filter). Approximately 10,000 events of a gated population were counted per sample.

Forward scatter (FSC) and side scatter (SSC) were recorded in a linear mode (in FSC vs. SSC dot plots). Data were acquired as FSC files using BD Cell Quest Pro v. 6.0, (Becton Dickinson Immunochemistry, San Jose, CA, USA). FlowJo^®^ Version 7.6.2 software (FlowJo^TM^, Ashland, OR, USA) was used to analyze the acquired data, using dot plots with the relative cell size (FSC), the internal complexity (SSC) and the specific fluorescence intensity for each probe.

#### 2.5.3. Viability

A LIVE/DEAD^®^ sperm viability kit was used to evaluate sperm viability and the evaluation was conducted along the lines recommended by Arando et al. [31]. A total of 100 μL of sperm was diluted with 150 μL of cytometer buffer to reach a final concentration ~4 × 10^6^ spz/mL. Then, 2.5 μL SYBR-14 (2 μM) and 5 μL PI (480 μM) were added and incubated in darkness conditions for 15 min. After incubation, the proportion of live/dead sperm cells was measured. Spermatozoa emitting in the green wavelength were deemed to be spermatozoa with intact plasma membranes and the results were reported as the percentage of spermatozoa with intact plasma membrane. Unstained and single-stained samples were used for calibrating the FSC gain, FL-1 and FL-2 PMT voltages and for compensation of SYBR-14 spill over into the PI channel (9.8%). Non-sperm particles (also called “alien events”) were located in the SYBR-14^−^/PI^−^ quadrant and they did not contain DNA. Spermatozoa with intact plasma membrane were located in the SYBR-14^+^/PI^−^ quadrant.

#### 2.5.4. Acrosome Integrity

Acrosome integrity was assessed using the combination of PNA-FITC and PI. One-hundred microliters of sperm (containing around 40 × 10^6^ spz) was incubated in darkness for 5 min with 5 μL of PNA-FITC stock solution (100 μg/mL in DMSO) and 5 μL PI (480 μM). After incubation, 400 μL of cytometer buffer was added and samples were analyzed. PI^−^ and PNA-FITC^−^ cells were categorized as sperm with intact acrosome and plasma membrane. Unstained and single-stained samples were used for setting the FSC gain, FL-1 and FL-2 PMT voltages and for compensation of PNA-FITC spill over into the PI channel (9.8%). The percentages of alien particles (f) determined by SYBR-14/PI staining were used to correct the percentages of non-stained spermatozoa (q1) in each sample in order to obtain the corrected percentage of non-stained spermatozoa (q1′), in accordance with Petrunkina & Harrison [32]:
q1′ = [(q1 − f)/(100 − f)] × 100.(1)

#### 2.5.5. Mitochondrial Membrane Potential

The combination of Mitotracker Red CMXRos and SYBR-14 was used to estimate mitochondrial membrane potential and it was assessed using a modified protocol [33,34]. A volume of 50 μL of sperm (containing 20 × 10^6^ spz) was mixed with 350 μL of cytometer buffer and immediately loaded with 2 μL SYBR-14 (2 μM) and 2 μL Mitotracker Red CMXRos (20 μM) was added. Sperm doses were incubated for 10 min at 37 °C in the dark and only sperm with high mitochondrial potential (HMMP) were reported. Unstained and single-stained samples were used for setting the FSC gain, FL-1 and FL-2 PMT voltages. Data were not compensated.

#### 2.5.6. Lipid Peroxidation

Lipid peroxidation (LPO) was estimated using C_11_-BODIPY^581/591^ (Molecular Probes Europe, Leiden, The Netherlands) using a modified protocol [35]. A volume of 100 μL of diluted sperm (containing around 2 × 10^6^ spz) was mixed with 1 μL C_11_-BODIPY^581/591^ (0.2 mM) and incubated at 37 °C for 30 min. After incubation, 1 mL of PBS was added for centrifugation at 600× *g* for 8 min. The pellet was resuspended with 100 μL of PBS prior to assessment. Spermatozoa with LPO emitted light in the green wavelength and were deemed to be BODIPY-positive cells. Unstained and single-stained samples were used for setting the FSC gain, FL-1 and FL-2 PMT voltages. Data were not compensated.

### 2.6. Data Analysis

Bayesian inference for ANOVA was run to test for statistical differences in the mean across antioxidants (HT, DHPG and MIX) at different concentrations (Control, HT1, HT2, HT3, HT4, DHPG1, DHPG2, DHPG3, DHPG4, MIX1, MIX2, MIX3 and MIX4) on buck sperm parameters (Table 1).

The Bayes factor (BF) quantifies the strength of the evidence of null and alternative hypotheses and is used instead of frequentist *p* values when Bayesian approaches are applied to issue conclusions. As BF increases, the degree at which evidence favors the alternative hypothesis compared to the null hypothesis increases as well. In this context, Cleophas and Zwinderman [36], suggested a method to extrapolate between the Bayes factor used in Bayesian approaches and *p* values from frequentist approaches to favor the interpretability of results. 

Sample descriptive posterior statistics are modeled from the means and variances of the measured unpaired groups and are provided as sources of variation, while the prior element was modeled as an uninformative prior using the Jeffreys–Zellener–Siow (JZS) method or, equivalently, from the computation of a reference prior based on a gamma distribution with a standard error of 1. As suggested by Martins-Bessa et al. [37], the 95% credibility interval shows that there is a 95% probability that these regression coefficients (posterior distribution mean value for each covariate and factor) in the population lie within the corresponding credibility intervals. When 0 is not contained in the credibility interval, a significant effect for such factor is detected. Integral calculation of factors is required for BF accuracy purposes. Therefore, afterwards, it can be used as a precise statistical index to measure the amount of support in favor of either H_1_ (the difference between the unpaired means is larger than zero) or H_0_ (the difference between the unpaired means is not larger than zero). Contextually, Bayesian approaches provide a better perspective of the structure model of the H_1_ and H_0_. Therefore, the maximal likelihoods of likelihood distributions are not always identical to the mean effect of traditional tests, which specifically fits the context of biological inferences, given biological likelihoods may better respond biological questions than numerical means of non-representative subgroups do.

IBM SPSS Statistics Algorithms version 25.0 by IBM Corp. [38] suggests Bayesian inference of ANOVA is approached as a special case of the Bayesian general multiple linear regression model. The algorithms used by SPSS to perform Bayesian Inference on Analysis of Variance (ANOVA) in this study are described in IBM SPSS Statistics Algorithms version 25.0 by IBM Corp. [38]. The tolerance value for the numerical methods and the number of method iterations were set as a default by SPSS v25.0 (IBM Corp., Armonk, NY, USA) [39].

The around 0 symmetric JZS prior was used as it is appropriate for Bayesian inference of ANOVA, provided positive and negative values of the slope parameters a priori have the same probability of occurring [40]. Furthermore, its scale-invariant properties, permits comparing parameters measured in different units, as it occurs in the present study. Bayesian inference for ANOVA was performed using the Bayesian Package of SPSS v25.0 (IBM Corp., Armonk, NY, USA) [39].

## 3. Results

### 3.1. Prior Descriptive Statistics 

The descriptive statistics of sperm viability, acrosome integrity, HMMP, and LPO parameters in fresh and frozen-thawed buck sperm are shown in the Table 1 and Table 2. 

### 3.2. Bayesian Inference of Olive Oil Derived Antioxidant Effect

Table 3 and Appendix A report the outputs from Bayesian ANOVA analysis and present posterior distribution statistics for sperm parameters across antioxidants and concentrations. Regarding to motility parameters, an evident/significant (*p* < 0.05) increase in PM was observed when HT2 treatment when compared to control treatment. However, at higher HT concentrations (70 µg/mL) an evident/significant (*p* < 0.05) decrease in PM occurred. 

When velocity parameters (VCL, VSL, and VAP) were evaluated, the mixture of both antioxidants produced a dose-dependent decrease in VSL and VAP values, with this being evidently/significantly (*p* < 0.05) lower when high concentrations were used. However, the rest of the studied kinetic parameters were not evidently/significantly affected (*p* > 0.05) by the addition of antioxidants, except for LIN when high concentrations of HT were added (70 µg/mL). 

Samples supplemented with low concentrations of HT and DHPG (10 µg/mL) reported evident/significant (*p* < 0.05) sperm viability increases in comparison to Control. This positive evident/significant (*p* < 0.05) effect was also observed for acrosome integrity in samples supplemented with low DHPG concentrations (10 µg/mL). By contrast, higher DHPG concentrations offered evidently lower acrosome integrity. Similarly, a negative effect was observed when high concentrations (70 µg/mL) of the mixture of both antioxidants were used. 

Mitochondrial potential evidently/significantly decreased (*p* < 0.05) in comparison to when the Control treatment was considered in samples supplemented with HT at low concentrations (10 and 30 µg/mL). Similarly, the addition of a mixture of both antioxidants evidently/significantly (*p* < 0.05) reduced the mitochondrial potential in frozen-thawed spermatozoa. Regarding to LPO values, the addition of 10 µg/mL of HT reported an evident/significant (*p* < 0.05) protective effect, reducing its value in comparison with Control treatments. However, when the dose increased, an opposite evident/significant (*p* < 0.05) effect was observed. By contrast, DHPG and MIX provided a dose-dependent better protection, diminishing LPO values as antioxidant concentration increases.

## 4. Discussion

Sperm cryopreservation offers goat breeders several benefits over fresh sperm storage. However, recent studies suggested that ROS concentration increases considerably during cryopreservation, disrupting sperm functions and subsequent fertilization [41]. Sometimes the endogenous antioxidant capacity of sperm cells is compromised due to the proliferation of ROS, producing an imbalance that promotes oxidative stress and consequently LPO, which affects membrane structure and distorts its functions such as membrane fluidity, membrane enzymes, ion gradients, receptor transduction, and transport processes [42]. In this context, the addition of antioxidants to the semen extender seems to have the potential to mitigate the negative impact of oxidative stress, as antioxidants capture free radicals and conclude the chain reaction, maintaining a redox state and offsetting their capacity to reduce molecular oxygen [43].

The addition of natural or synthetic antioxidants to the cryopreservation medium in goat sperm has attracted the attention of researchers as an alternative to diminish the negative effect of oxidative stress produced by ROS and to improve post-thawed sperm quality [11,44,45]. However, it is not easy to prove the exact nature of the action of antioxidants on sperm quality, and the degree to which interactions with other factors such as the species, the extender and the type or the concentration of antioxidant may be involved [46]. There is a knowledge gap on whether antioxidants are absorbed unchanged or metabolized into completely different compounds. Furthermore, the efficacy of the common antioxidants, such as vitamins C and E, selenium, and herbal supplements to reduce pathological ROS has not yet been determined [47]. 

In this context, HT is soluble in both lipid and water solutions, and therefore soluble in all phases of the heterogeneous system studied in the present research. The concentration of HT in biological systems is very similar in both aqueous and lipid areas [48]. As far as DHPG is concerned, this component has only recently been isolated and there is relatively little information regarding the way it behaves. On the basis of its chemical structure, which is very similar to HT, it should be broadly analogous to HT, and it is also soluble in both lipid and water solutions.

To the present authors’ knowledge, this is the first study in which HT and DHPG were tested as antioxidants for caprine sperm cryopreservation. Biological activity and risk/benefit of polyphenolic compounds are dependent on their diversity, dual-effects, biological activity, and source [49]. In this regard, the effect of olive oil and olive oil-derived antioxidants on sperm quality has previously been investigated in other species. In rabbits, olive oil administered at 7% *v*/*w* for 16 weeks succeeded in recovering the loss of volume, count, motility, and normal spermatozoa in males exposed to a hypercholesterolemic diet [50]. Similarly, the oral administration of olive oil in healthy rats at 0.4 mL daily for six weeks improved the sperm parameters [51]. Banihani [52] concluded that the addition of olive oil preserves semen quality by enhancing the gonadal function, reducing oxidative injury and lipid peroxidation, and promoting nitric oxide signaling.

Results derived from the present study reported the fact that in samples supplemented with 30 µg/mL of HT an increase in progressive motility of 11% was reported in comparison to Control treatment which agrees the results reported by Hamden et al. [22], who supplemented rat sperm with 50 µg/mL HT and those by Krishnappa et al. [24] who reported an improvement in total motility when 80 mM HT was added to ovine sperm. These findings suggest that the presence of HT could mitigate ROS concentration, preventing the negative impact of moderately elevated ROS concentrations on the sperm movement, mostly via depletion of intracellular ATP and the successive reduction in the phosphorylation of axonemal proteins [53]. However, in the present study, PM considerably decreased when high concentrations (70 µg/mL) of HT were added. One of the reasons for this negative effect associated to the increase of the dose would be the extender acidification, as previously described by Ibrahim et al. [54], who supplemented goat sperm with alpha lipoic acid. By contrast, no effect was observed in the present study for TM and PM when DHPG and MIX were added, in line with previous studies carried out on sheep using the same antioxidants and concentrations [15,25] and in incubated human sperm [23] after HT supplementation. 

In regard to kinematic parameters, no effect was observed when HT and DHPG were independently added. However, the mixture of both antioxidants induced a decrease of VSL and VAP in dose-dependent manner, being ~13% when high concentrations were used. This similar trend was previously described by Arando, et al. [15] in liquid ram sperm stored at 5 °C, suggesting that high concentrations of these antioxidants could be deleterious for spermatozoa. Contextually, broad evidence suggests higher concentrations may not necessarily translate into better quality, but indeed may be detrimental [55]. In agreement with the present study, the use of other antioxidants such as arbutin, butylated hydroxyanisole, rosemary or lycopene, have reported a positive effect on goat sperm motility [11,44,45,56].

A high amount of PUFA in sperm membranes could interact with ROS, affecting membrane fluidity, facilitating Ca^2+^ influx, and provoking membrane protein reorganization and the destabilization of the plasma membrane [57]. Based on the current results, the addition of HT at low concentrations (10 µg/mL) may improve membrane integrity, showing an increase of 14% compared to Control treatment, as reported by Hamden et al. [22] when HT was added. Similarly, some authors reported an increase of membrane integrity when other antioxidants were added in extender medium [8,56,58,59].

Acrosome is a specialized sperm structure comprising membranes and proteins which makes it highly susceptible to ROS-derived damage [60]. In this context, a positive effect was observed in acrosome integrity when samples were supplemented with low DHPG concentrations (10 µg/mL). By contrast, as DHPG concentration increases or when high concentrations of mixture (70 µg/mL) were used, acrosome integrity decreased, as previously described by Arando et al. [25]. On the other hand, Hashem et al. [61] reported higher acrosome integrity values when oleic acid was added to ram sperm. Similarly, recent studies in buck sperm showed that the addition of different antioxidants, as vitamin C or lycopene, could mitigate the acrosome damage [11,62].

Mitochondria are involved in the generation of ROS in spermatozoa through the pathway of nicotinamide adenine dinucleotide-dependent oxide reductase reactions, which directly affects their normal functions [63]. Endogenous antioxidant supplementation has bene hypothesized to reduce oxidative stress and, as a consequence, to maintain post-thawed mitochondrial potential. In this sense, the present study reveals that mitochondrial membrane potential is not improved in freezing-thawing sperm samples supplemented with HT, DHPG, and/or MIX, in comparison to the Control treatment, as reported by Zanganeh et al. [45] and Arando et al. [25].

However, in a recent study using mitochondria-targeted antioxidants, the authors noted a slight significant mitochondrial potential increase [64]. The olive-oil antioxidants tested in the present study did not offer any advantage for the mitochondrial activity after cryopreservation. However, further studies should be conducted to elucidate why the intense impact of freezing-thawing process on the mitochondrial activity in buck sperm cannot be counteract by the mentioned antioxidants. By contrast, other authors observed a significant increase in mitochondrial potential when cysteine, coenzyme Q_10,_ lycopene or alpha-lipoic acid were used [10,11,65].

As far as LPO is concerned, the present study showed that the addition of 10 µg/mL of HT may present a protective function against lipid peroxidation, reducing it by 19% in comparison to Control group. In reference to oil-derived antioxidants, a recent study also reported that the addition of HT and DHPG improved LPO values in ram sperm [25]. However, the opposite effect was observed as dose increased. DHPG and the mixture of both antioxidants provided better protection properties, diminishing LPO values around 20%, when sperm samples were supplemented with high doses of antioxidants.

## 5. Conclusions

HT could mitigate ROS concentration, preventing their negative impact on the sperm movement. Dose-dependent extender acidification produces a considerable reduction of PM. The mixture of both antioxidants induced a decrease of VSL and VAP in dose dependent manner, being approximately 13% when high concentrations were used. Higher concentrations may not necessarily lead to better results but may be detrimental for sperm quality. The addition of HT at low concentrations (10 µg/mL) may improve membrane integrity. Acrosome integrity improves when supplementing with low DHPG concentrations (10 µg/mL). By contrast, as DHPG concentration increases or when high concentrations of mixture (70 µg/mL) were used, acrosome integrity decreased. Mitochondrial membrane potential is not improved in freezing-thawing sperm supplemented with HT, DHPG and/or MIX. Olive-oil antioxidants did not benefit mitochondrial activity after cryopreservation. The addition of 10 µg/mL of HT may present a protective function against lipid peroxidation, reducing it by 19%. However, the opposite effect was observed as dose increased. DHPG and the mixture of both antioxidants provided better protection properties, diminishing LPO values around 20%, when sperm was supplemented with high doses of antioxidants. Therefore, in light of the obtained results the addition of HT at 10 µg/mL and DHPG at 30 µg/mL were the most suitable treatments since they improved post-thawing sperm quality.

## Figures and Tables

**Figure 1 animals-11-02032-f001:**
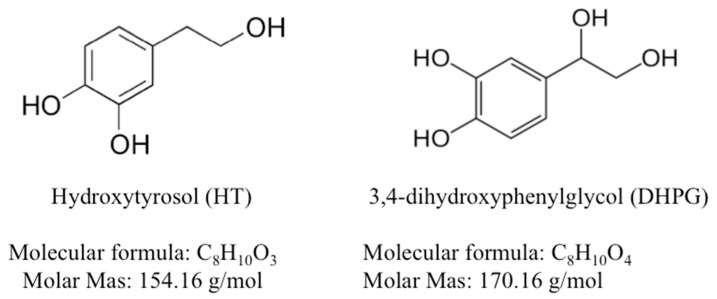
Chemical structure of olive oil-derived antioxidants used in the present study.

**Figure 2 animals-11-02032-f002:**
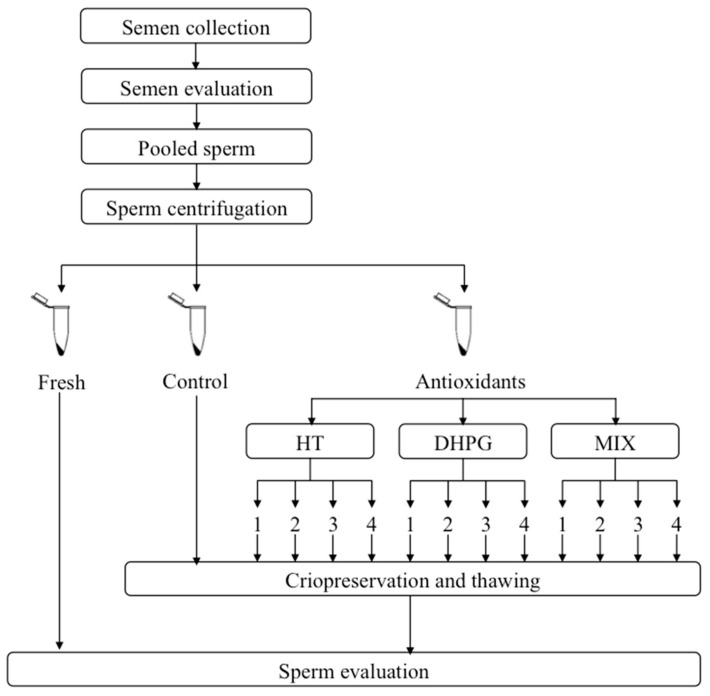
Schematic overview of the experimental design.

**Table 1 animals-11-02032-t001:** Descriptive statistics of sperm viability, acrosome integrity, HMMP, and LPO parameters in fresh and frozen-thawed buck sperm. Data are expressed as Mean ± SD.

	Viability (%)	Acrosome Integrity (%)	HMMP (%)	LPO (%)
FRESH	84.4 ± 4.5	72.0 ± 6.4	79.8 ± 7.1	1.9 ± 0.8
CONTROL	43.9 ± 7.2	43.0 ± 7.7	40.3 ± 6.6	2.1 ± 0.5
HT1	50.2 ± 6.5	40.2 ± 9.8	32.8 ± 4.4	1.7 ± 0.6
HT2	44.5 ± 5.9	37.7 ± 5.5	32.6 ± 7.0	2.2 ± 0.4
HT3	44.5 ± 8.8	42.2 ± 11.4	38.2 ± 5.6	2.5 ± 0.7
HT4	43.1 ± 9.0	42.3 ± 6.9	40.3 ± 7.1	2.2 ± 0.8
DHPG1	49.9 ± 14.2	47.5 ± 11.3	40.8 ± 9.0	2.2 ± 1.3
DHPG2	43.8 ± 6.7	41.7 ± 6.4	35.8 ± 8.2	1.6 ± 0.5
DHPG3	43.6 ± 7.0	38.8 ± 6.2	36.8 ± 9.1	1.8 ± 0.9
DHPG4	47.0 ± 8.5	35.0 ± 8.3	40.0 ± 6.0	1.3 ± 0.7
MIX1	43.4 ± 7.9	39.0 ± 5.2	35.8 ± 7.3	2.9 ± 1.5
MIX2	44.2 ± 8.6	39.9 ± 7.9	36.5 ± 7.3	1.6 ± 0.7
MIX3	42.6 ± 13.3	41.7 ± 12.7	38.0 ± 3.1	1.6 ± 0.6
MIX4	43.2 ± 8.5	38.8 ± 8.9	34.0 ± 8.1	1.6 ± 0.7

HMMP: mitochondrial potential; LPO: lipid peroxidation; HT: hydroxytyrosol; DHPG: 3,4-dihydroxyphenylglycol; CONTROL (without antioxidant); HT1 (10 μg/mL); HT2 (30 μg/mL); HT3 (50 μg/mL) and HT4 (70 μg/mL); DHPG1 (10 μg/mL); DHPG2 (30 μg/mL); DHPG3 (50 μg/mL); DHPG4 (70 μg/mL); MIX1 (5 μg/mL HT + 5 μg/mL DHPG); MIX2 (15 μg/mL HT + 15 μg/mL DHPG); MIX3 (25 μg/mL HT + 25 μg/mL DHPG); MIX4 (35 μg/mL HT + 35 μg/mL DHPG).

**Table 2 animals-11-02032-t002:** Descriptive statistics of sperm motility and kinematic parameters in fresh and frozen-thawed buck sperm.

	MT (%)	PM (%)	VLC (μm/s)	VSL (μm/s)	VAP (%)	LIN (%)	STR (%)	WOB (%)	ALH (μm)	BFC (Hz)
FRESH	91.6 ± 3.4	38.9 ± 4.8	105.4 ± 9.4	36.9 ± 4.9	64.3 ± 4.3	35.2 ± 4.7	57.4 ± 5.7	61.1 ± 3.3	3.3 ± 0.4	10.1 ± 1.1
CONTROL	60.2 ± 9.4	31.6 ± 6.5	82.5 ± 14.3	39.7 ± 8.9	54.5 ± 11.7	47.7 ± 3.8	72.7 ± 2.2	65.7 ± 4.0	3.0 ± 0.3	11.0 ± 0.6
HT1	55.7 ± 6.4	30 ± 8.5	78.1 ± 10.0	37.6 ± 7.8	51.2 ± 9.0	48.0 ± 7.1	73.1 ± 4.7	65.4 ± 6.6	2.8 ± 0.3	11.4 ± 1.0
HT2	60.0 ± 8.9	35.2 ± 5.3	79.7 ± 6.8	38.9 ± 4.9	52.3 ± 5.2	49.0 ± 6.7	74.3 ± 4.2	65.8 ± 6.3	2.9 ± 0.4	11.2 ± 1.0
HT3	63.7 ± 8.1	33.9 ± 5.8	77.0 ± 10.2	36.1 ± 6.4	49.6 ± 9.1	46.9 ± 5.6	73.0 ± 3.2	64.2 ± 6.6	2.9 ± 0.3	11.1 ± 1.2
HT4	54.5 ± 7.9	27.7 ± 8.2	85.7 ± 9.2	36.2 ± 4.6	52.9 ± 6.8	42.2 ± 2.9	68.5 ± 3.1	61.6 ± 2.6	3.0 ± 0.2	11.9 ± 0.2
DHPG1	58.9 ± 9.8	32.8 ± 8	83.7 ± 10.8	37.6 ± 3.3	52.8 ± 5.4	45.3 ± 3.9	71.4 ± 2.3	63.3 ± 3.8	3.1 ± 0.3	12.1 ± 0.9
DHPG2	55.5 ± 9.8	31.8 ± 4.7	85.0 ± 10.0	39.8 ± 5.6	55.1 ± 8.3	46.8 ± 3.2	72.5 ± 2.3	64.6 ± 3.7	2.9 ± 0.2	11.8 ± 0.4
DHPG3	61.6 ± 9.1	32.4 ± 7.9	78.5 ± 12.4	35.2 ± 5.1	49.0 ± 6.2	45.2 ± 4.9	71.9 ± 4.4	62.7 ± 3.4	3.0 ± 0.3	11.6 ± 1.0
DHPG4	59.7 ± 9.8	33.9 ± 4.4	84.9 ± 8.5	37.9 ± 3.6	52.5 ± 4.7	44.7 ± 3.1	72.2 ± 2.7	62.0 ± 4.0	3.1 ± 0.3	12.2 ± 1.1
MIX1	59.8 ± 9.4	30.7 ± 8.3	81.0 ± 10.7	36.3 ± 6.0	50.7 ± 6.4	44.9 ± 4.3	71.4 ± 4.1	62.8 ± 2.9	3.0 ± 0.2	11.4 ± 1.3
MIX2	60.0 ± 6.6	31.5 ± 7.8	80.1 ± 6.6	35.3 ± 4.0	49.6 ± 3.5	44.1 ± 2.8	71.1 ± 3.5	62.0 ± 1.6	3.1 ± 0.2	11.4 ± 0.9
MIX3	57.4 ± 9.0	32.3 ± 9.1	76.8 ± 5.5	34.4 ± 3.1	47.2 ± 4.2	44.8 ± 3.5	72.9 ± 5.0	61.4 ± 2.0	3.0 ± 0.2	11.7 ± 1.0
MIX4	58.1 ± 8.1	31.4 ± 3.6	77.3 ± 5.5	34.5 ± 2.6	47.5 ± 3.9	44.6 ± 2.0	72.7 ± 1.9	61.4 ± 2.8	3.1 ± 0.1	11.9 ± 0.7

TM: Total motility; PM: progressive motility, VCL: curvilinear velocity; VSL:, straight line velocity; VAP: average path velocity; STR: straightness; LIN: linearity; WOB: wobble; ALH: amplitude of lateral head displacement; BCF: beat/cross frequency; HT: hydroxytyrosol; DHPG: 3,4-dihydroxyphenylglycol; CONTROL (without antioxidant); HT1 (10 μg/mL); HT2 (30 μg/mL); HT3 (50 μg/mL) and HT4 (70 μg/mL); DHPG1 (10 μg/mL); DHPG2 (30 μg/mL); DHPG3 (50 μg/mL); DHPG4 (70 μg/mL); MIX1 (5 μg/mL HT + 5 μg/mL DHPG); MIX2 (15 μg/mL HT + 15 μg/mL DHPG); MIX3 (25 μg/mL HT + 25 μg/mL DHPG); MIX4 (35 μg/mL HT + 35 μg/mL DHPG).

**Table 3 animals-11-02032-t003:** Summary of the outputs of Bayesian inference ANOVA to detect differences in the mean of sperm parameters across the different concentrations of HT, DHPG, and MIX antioxidants.

	BG Sum of Squares	BG df	BG Mean Square	WG Sum of Squares	WG df	WG Mean Square	F	*p* Value	Bayes Factor
Viability (%)	9146.527	12	703.579	5301.865	70	75.741	9.289	0.001	14,617,034.109
Acrosome integrity (%)	6166.972	12	474.382	5044.732	70	72.068	6.582	0.001	11,102.852
HMMP (%)	10,748.051	12	826.773	3465.358	70	49.505	16.701	0.001	24,196,033,653,804.700
LPO (%)	13.991	12	1.076	48.149	70	0.688	1.565	0.117	0.000
TM (%)	6445.744	12	495.826	4985.240	70	71.218	6.962	0.001	33,431.056
PM (%)	530.067	12	40.774	3293.809	70	47.054	0.867	0.591	0.000
VCL (μm/s)	4134.502	12	318.039	6422.702	70	91.753	3.466	0.001	0.252
VSL (μm/s)	247.939	12	19.072	1998.821	70	28.555	0.668	0.787	0.000
VAP (μm/s)	1402.105	12	107.854	3178.839	70	45.412	2.375	0.011	0.002
LIN (%)	869.610	12	66.893	1361.286	70	19.447	3.440	0.001	0.227
STR (%)	1344.017	12	103.386	960.978	70	13.728	7.531	0.001	163,617.988
WOB (%)	222.771	12	17.136	1195.420	70	17.077	1.003	0.458	0.000
ALH (μm)	0.842	12	0.065	4.724	70	0.067	0.960	0.499	0.000
BCF (Hz)	23.095	12	1.777	60.923	70	0.870	2.041	0.029	0.000

HMMP: mitochondrial potential; LPO: lipid peroxidation; BG: Between groups; df: degrees of freedom; WG: Within groups; TM: Total motility; PM: progressive motility, VCL: curvilinear velocity; VSL: straight line velocity; VAP: average path velocity; STR: straightness; LIN: linearity; WOB: wobble; ALH: amplitude of lateral head displacement; BCF: beat/cross frequency.

## Data Availability

Data will be made accessible from corresponding authors upon reasonable request.

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
