# Peer review of "Bayesian Analysis of the Effects of Olive Oil-Derived Antioxidants on Cryopreserved Buck Sperm Parameters"

_animals, 2021, doi:10.3390/ani11072032_

Round 1

Reviewer 1 Report

The authors revised the original manuscript. Replied to the comments. The manuscript can be published 

Author Response

Response: We thank the reviewer for his/her attention to our manuscript.

Reviewer 2 Report

The author's response to my previous suggestions seemed well justified to me, although they did not say anything about the defrost system. Although there are various seminal thawing methods in the literature, the one that has worked best for us is the one we suggest. Therefore, we would recommend that you use it and compare it.

Author Response

The author's response to my previous suggestions seemed well justified to me, although they did not say anything about the defrost system. Although there are various seminal thawing methods in the literature, the one that has worked best for us is the one we suggest. Therefore, we would recommend that you use it and compare it.

Response: We understand and appreciate the reviewer suggestion and of course we will borrow the idea and take it in account for future studies. At this point it is unfeasible to perform this for this study and may not add much to the aim of the paper which was to evaluate the effect of antioxidants. The decision to perform the method described in the paper was made upon the fact that several thawing methods and exposure times have been reported to lead to non-conclusive results. As a result, we consulted other recent references for studies in goat sperm, and determined this may be one of the most common procedures to implement (37ºC and 30 s) [1, 2, 3, 4].

  • Alcay, S., Ustuner, B., Aktar, A., Mulkpinar, E., Duman, M., Akkasoglu, M., & Cetinkaya, M. (2020). Goat semen cryopreservation with rainbow trout seminal plasma supplemented lecithin‐based extenders. Andrologia, 52(4), e13555.
  • Tabarez, A., García, W., & Palomo, M. J. (2017). Effect of the type of egg yolk, removal of seminal plasma and donor age on buck sperm cryopreservation. Small Ruminant Research, 149, 91-98.
  • Ren, F., Feng, T., Dai, G., Wang, Y., Zhu, H., & Hu, J. (2018). Lycopene and alpha-lipoic acid improve semen antioxidant enzymes activity and cashmere goat sperm function after cryopreservation. Cryobiology, 84, 27-32.
  • Fathi, M., Zaher, R., Ragab, D., Gamal, I., Mohamed, A., Abu-El Naga, E., & Badr, M. (2019). Soybean lecithin–based extender improves Damascus goat sperm cryopreservation and fertilizing potential following artificial insemination. Asian Pacific Journal of Reproduction, 8(4), 174.

Reviewer 3 Report

Line 126-129; 297-298: Change commas to semicolons.

Line 295-296: Table 1. Add the word fresh (parameters in fresh and frozen-thawed buck sperm) or make a table with the data of the fresh semen.

Line 298: Table 2. Add the word fresh (kinematic parameters in fresh and frozen-thawed buck sperm) or make a table with the data of the fresh semen.

Author Response

Reviewer 3:

Line 126-129; 297-298: Change commas to semicolons.

Response: Changed as suggested by the reviewer.

Line 295-296: Table 1. Add the word fresh (parameters in fresh and frozen-thawed buck sperm) or make a table with the data of the fresh semen.

Response: Changed as suggested by the reviewer.

Line 298: Table 2. Add the word fresh (kinematic parameters in fresh and frozen-thawed buck sperm) or make a table with the data of the fresh semen.

Response: Changed as suggested by the reviewer.

Reviewer 4 Report

The aim of this paper was essentially to evaluate the effect of olive oil-derived antioxidants on cryopreserved caprine sperm, by using a Bayesian analysis.

The description of the results is not well structured and clear.

In my opinion the impact and the novelty of the research is too low to be published in Animals.

Author Response

Reviewer 4:

The aim of this paper was essentially to evaluate the effect of olive oil-derived antioxidants on cryopreserved caprine sperm, by using a Bayesian analysis.

The description of the results is not well structured and clear.

Response: Result section was revised and restructured to make the presentation of the outputs of the study clearer and more readable.

In my opinion the impact and the novelty of the research is too low to be published in Animals.

Response: We disagree with the reviewer comments. Up to our knowledge no study of the kind (antioxidant derived from olive oil) has been performed in goats. Furthermore, this may be the first time that Bayesian methodologies have been used to evaluate data of this kind. In this context, first quartile ranked journals in the area have published papers dealing with the same topic (not on the same species or using the same statistical approach).

Arando, A., Delgado, J. V., Bermúdez‐Oria, A., León, J. M., Fernández‐Prior, Á., Nogales, S., & Pérez‐Marín, C. C. (2020). Effect of olive‐derived antioxidants (3, 4‐dihydroxyphenylethanol and 3, 4 dihydroxyphenylglycol) on sperm motility and fertility in liquid ram sperm stored at 15° C or 5° C. Reproduction in Domestic Animals55(3), 325-332.

Arando, A., Delgado, J. V., Fernández-Prior, A., León, J. M., Bermúdez-Oria, A., Nogales, S., & Pérez-Marín, C. C. (2019). Effect of different olive oil-derived antioxidants (hydroxytyrosol and 3, 4-dihydroxyphenylglycol) on the quality of frozen-thawed ram sperm. Cryobiology, 86, 33-39.

This manuscript is a resubmission of an earlier submission. The following is a list of the peer review reports and author responses from that submission.

Round 1

Reviewer 1 Report

This is a very interesting research about the effects of olive oil-derived antioxidants on cryopreserved buck sperm parameters. The experimental design is according with the objective proposed, and the results obtained are clearly exposed and discussed.

Comments:

  1. Please show more information that is mass motility. How do you evaluate it?Which means score "4"?
  2. Is there any data on artificial insemination?
  3.  No conclusion is it recommended or not to add olive oil-derived antioxidants to the extender and and their suitable  amount. Please add. 

Author Response

Reviewer 1:

This is a very interesting research about the effects of olive oil-derived antioxidants on cryopreserved buck sperm parameters. The experimental design is according with the objective proposed, and the results obtained are clearly exposed and discussed.

Response: We thank the reviewer for his/her kind comment.

  • Please show more information that is mass motility. How do you evaluate it? Which means score "4"?

Response: We added the information requested to the body text “The volume was quantified by micropipetting and mass motility, assessed by placing a drop of 5 μL of pure semen on a slide preheated at 36°C in the microscopic, heated stage and observed under an optical microscope at a magnification of 100 x in several microscopic fields, This mass sperm motility was scored subjectively from 0 (no motion) to 5 (numerous rapid waves) on a scale with steps equal to 1 according to the original method described by Evans and Maxwell [28] and Lopes, et al. [29] Only ejaculates with more than 70% motility (score 4 and 5) were evaluated and cryopreserved.” Citations included were adapted accordingly.

  • Is there any data on artificial insemination?

Response: No, unfortunately we have not done any artificial insemination trial. This was the first attempt where the effect of hydroxytyrosol (HT) and 3,4-dihydroxyphenylglycol (DHPG) was tested in criopreservated buck sperm. In this sense, authors considered that as first step it was necessary to assess in vitro behaviour of these natural olive oil derived antioxidant in order to determine their real possibilities to use in in vivo tests.  In addition, it is necessary to consider that in vivo studies are conducted in commercial herds, so we need to be sure that we do not create any setbacks, which result in economic damage to farmers. However, taking into account the results, authors considered for further studies the possibility to make in vivo studies where the real potential of these antioxidants will be tested.

  • No conclusion is it recommended or not to add olive oil-derived antioxidants to the extender and and their suitable amount. Please add. 

Response: As the reviewer 1 suggests, the requested information has been added as a conclusion of the study.

Reviewer 2 Report

This study has great scientific value in the world of goat production, especially the importance of obtaining frozen semen. This technique still has big drawbacks, especially in local breeds. In nature are many antioxidants presented by vegetable world that they play a important role in cell metabolism that makes them more resistant to external adversities.

In the caprine sperm the outer and internal membranes consist of a large number of phospholipids susceptible to attack by ROS and subjected to oxidative stress. This work test two  of these types of natural antioxidants and shows the quantities and concentrations al which these substances act effectively on the sperm cell.

Preliminary study, methodology and materials used are very suitable for achievement of results. The design of the test and statistical methodology is appropriate in addiction to novel in this field. The exposure of the results is clearly expressed and provides a good tool for further discussion an conclusions, and the literature references are appropriate to this study.

Only need a small grammatical review in the table holders presented.

This work needs to be completed with studies in live animals to understand the efficiency and effectiveness of artificial insemination with thawed semen.

Author Response

Reviewer 2:

This study has great scientific value in the world of goat production, especially the importance of obtaining frozen semen. This technique still has big drawbacks, especially in local breeds. In nature are many antioxidants presented by vegetable world that they play a important role in cell metabolism that makes them more resistant to external adversities.

Response: We thank the reviewer for his/her kind comment.

In the caprine sperm the outer and internal membranes consist of a large number of phospholipids susceptible to attack by ROS and subjected to oxidative stress. This work test two  of these types of natural antioxidants and shows the quantities and concentrations al which these substances act effectively on the sperm cell.

Preliminary study, methodology and materials used are very suitable for achievement of results. The design of the test and statistical methodology is appropriate in addiction to novel in this field. The exposure of the results is clearly expressed and provides a good tool for further discussion and conclusions, and the literature references are appropriate to this study.

  • Only need a small grammatical review in the table holders presented.

Response: Table holders were revised as suggested by the reviewer.

  • This work needs to be completed with studies in live animals to understand the efficiency and effectiveness of artificial insemination with thawed semen.

Response: Although unfortunately, we have not done any artificial insemination trial, we really appreciate and will consider this for future studies. This was the first attempt where the effect of hydroxytyrosol (HT) and 3,4-dihydroxyphenylglycol (DHPG) was tested in criopreservated buck sperm. In this sense, authors considered that as first step it was necessary to assess in vitro behaviour of these natural olive oil derived antioxidant in order to determine their real possibilities to use in in vivo tests.  It is necessary to consider that in vivo studies are conducted in commercial herds, so we need to be sure that we do not create any setbacks, which result in economic damage to farmers. However, taking into account the results, authors considered for further studies the possibility to make an in vivo studies were the real potential of these antioxidants will be tested.

Reviewer 3 Report

In the sperm freezing process I recommend using a cryocooler and not liquid nitrogen vapors. The use of a cryo-freezer makes the results obtained more objective and extrapolated.

I also advise using another way to defrost the straws, which would defrost faster. In our Equipment we have been using a defrost at 56ºC for 12 seconds for quite some time.

Author Response

Reviewer 3:

  • In the sperm freezing process I recommend using a cryocooler and not liquid nitrogen vapors. The use of a cryo-freezer makes the results obtained more objective and extrapolated. I also advise using another way to defrost the straws, which would defrost faster. In our Equipment we have been using a defrost at 56ºC for 12 seconds for quite some time.

Response: We deeply appreciate the suggestion made and we take it in account for future studies. However, when the study was conducted a cryo-freezer was not available. That is why we considered the use of nitrogen vapours. We agree with the reviewer on the fact that the use of the cryo-freezer may offer more objective and extrapolated results; however, the use of nitrogen vapours is highly extended in domestic animal sperm cryopreservation and in buck sperm as can be seen in recent studies [1, 2, 3, 4]. .

  • Naderi, N., Hajian, M., Souri, M., Esfahani, M. H. N., & Vash, N. T. (2021). Ferulago angulata extract improves the quality of buck spermatozoa post-thaw and counteracts the harmful effects of diazinon and lead. Cryobiology, 98, 17-24.
  • Azimi, G., Farshad, A., Farzinpour, A., Rostamzadeh, J., & Sharafi, M. (2020). Evaluation of used Purslane extracts in Tris extenders on cryopreserved goat sperm. Cryobiology, 94, 40-48.
  • Tabarez, A., García, W., & Palomo, M. J. (2020). Soy lecithin as a potential alternative to powdered egg yolk for buck sperm cryopreservation does not protect them from mitochondrial damage. Animal reproduction science, 217, 106473.
  • Flores-Gil, V. N., Toledano-Díaz, A., Velázquez, R., Oteo, M., López-Sebastián, A., & Santiago-Moreno, J. (2021). Role of changes in plasma prolactin concentrations on ram and buck sperm cryoresistance. Domestic Animal Endocrinology, 76, 106624.

Reviewer 4 Report

The manuscript is not suitable for publication. The main problems are the experimental design and the data presentation.

Regarding the experimental design, it is not clear how many samples are in each group. 

The presentation of the data as mean+/- ds is not adequate for a scientific manuscript. P values must be included in the tables and would be better to present a general source of variation. No footnote are in the tables

Author Response

Reviewer 4:

The manuscript is not suitable for publication. The main problems are the experimental design and the data presentation.

  • Regarding the experimental design, it is not clear how many samples are in each group. 

Response: The experimental information requested by the reviewer was included in the body text.

  • The presentation of the data as mean+/- ds is not adequate for a scientific manuscript. P values must be included in the tables and would be better to present a general source of variation.

Response: We agree with the reviewer and this information was just informative for readers. Anyway, Supplementary Table 1 included all posterior distribution Bayesian descriptive statistics, hence we think all sources of potential variability were covered. Due to the enormous extension of this material we decided to include it as a supplementary source. In regards, p values, the Bayes factor (BF) measures the likelihood of null and alternative hypotheses or one model versus another based on the prior distribution and the data. It quantifies the change in the likelihood given in the prior to the posterior likelihood that is produced by the data. The BF is a measure of the strength of the evidence and is used instead of p values (from frequentist approaches) to reach a conclusion. A large BF implies that the evidence favours the alternative hypothesis compared to the null hypothesis, or of one model over the other. A BF of 10, for instance, implies that the model comprising an intercept and a certain combination of factors is 10 times more likely than the comparison model, that is a model comprising the intercept and no other factor. Bayes factors can be used for any pair of models. A BF larger than 10 may be considered strong or very strong evidence for that model while very small values strongly favour the null hypothesis or the model comprising the intercept, but there is no generally accepted standard. Commonly used thresholds to define significance of evidence following the premises by Jeffreys [1] and Lee and Wagenmakers [2]. Still, as suggested by Cleophas and Zwinderman [3] extrapolation between the Bayes factor used in Bayesian approaches and p values from frequentist approaches could be performed to favor the interpretability of results. Hence, we included p value extrapolation, as p values themselves do not exist in Bayesian inference.

  1. Jeffreys, H. Theory of Probability, 3rd Edn Oxford: Oxford University Press. 1961.
  2. Lee, M.; Wagenmakers, E. Bayesian data analysis for cognitive science: A practical course. 2013.
  3. Cleophas, T.J.; Zwinderman, A.H. Modern bayesian statistics in clinical research; Springer: 2018.

No footnote are in the tables

Response: Footnotes were enclosed.